# Assessment of the Correlation between the Levels of Physical Activity and Technology Usage among Children with Down Syndrome in the Riyadh Region

**DOI:** 10.3390/ijerph191710958

**Published:** 2022-09-02

**Authors:** Reem. M. Alwhaibi, Asma B. Omer, Ruqaiyah Khan, Felwa Albashir, Noura Alkuait, Rawan Alhazmi

**Affiliations:** 1Department of Rehabilitation Sciences, College of Health and Rehabilitation Sciences, Princess Nourah Bint Abdulrahman University, P.O. Box 84428, Riyadh 11671, Saudi Arabia; 2Department of Basic Health Sciences, Deanship of Preparatory Year for the Health Colleges, Princess Nourah Bint Abdulrahman University, P.O. Box 84428, Riyadh 11671, Saudi Arabia; 3Department of Public Health, Athar Institute of Health and Management Studies, New Delhi 110049, India

**Keywords:** down syndrome, children, physical activity, technology, electronic device

## Abstract

**Background**: Children with Down Syndrome (C-DS) have language, cognitive and communication difficulties, in addition to consistent physical inactivity that contributes to poor health and higher-disability-adjusted life years. The purpose of this study was to determine the correlation between the use of electronic technology and levels of physical activity in C-DS in the Riyadh region of Saudi Arabia. **Methods**: A cross-sectional study was conducted with 49 mothers, where each had a child (6–12 years of age) with Down Syndrome (DS), and who were recruited using purposive sampling from three DS centers in Riyadh, Saudi Arabia. The Children’s Physical Activity Questionnaire and Research Questionnaire on the Impact of Technology on Children were used. Descriptive statistics were used to describe the demographics. Pearson’s correlation, Student’s *t*-test and the Chi-square test were used to assess the association between technology use, physical activity levels and socio-demographic variables. **Results**: There was no significant correlation between physical activity and the use of technology by C-DS. However, there was a negative correlation between a high level of physical activity and technology use (R = −0.037). Although, no significant correlation between the mother’s characteristics and technology use was found; there was a significantly positive correlation (*p* = 0.05) between the education level of mothers and the technology use by C-DS. Nonetheless, there was no association between the physical activity level and the gender of the child with DS. **Conclusions**: This study found that no significant relationship exists between the use of electronic gadgets and the level of physical activity in C-DS.

## 1. Introduction

Children suffering from Down Syndrome (DS) have additional genetic material from chromosome 21, which results in multiple deformities, intellectual impairment, and the presence of certain medical conditions [1]. DS is one of the world’s most-frequent chromosomal diseases. Every year, around 6000 newborns in the United States are born with DS [2]. In United Kingdom, almost 750 newborns with DS [3] are reported every year. In 2015, a large-scale study by AlSalloum et.al, which assessed the prevalence of congenital abnormalities amongst 45,682 children in Saudi Arabia has reported that DS was the most frequent with a prevalence rate of 6.6/10,000 [4].

C-DS experience communication, cognitive and verbal difficulties, as well as physical inactivity that negatively impact their quality of life [5,6]. They present with numerous physical findings such as small chin, slanted eye, poor muscle tone, a flat nasal bridge, a single crease of the palm and a protruding due to small mouth and large tongue, that are relatively variable from one child to another. Nevertheless, intellectual impairment, of which the severity can range from mild to severe, is present in all cases [7]. DS have a significant risk of developing certain medical conditions, such as Alzheimer’s disease. Additionally, they may also develop ophthalmic problems, including cataracts, with severe refractive errors occurring in approximately 60% of cases and around 75% of DS cases are associated with hearing loss. Furthermore, an estimated 40% of DS babies are born with congenital heart disease, and less commonly, thyroid disease and leukemia [7]. In addition to the health issues discussed above, C-DS exhibit a set of psycho-social, cognitive, anatomical and physiological attributes which predispose them to limitations in physical activity and physical fitness [1].

Juvenile obesity is an increasingly common health concern for C-DS; but their increased indulgence in moderate-to-vigorous intensity physical activity (MVPA) could assist in preventing obesity and enhance lifelong health for C-DS [8]. WHO suggested that all children should get at least one hour of MVPA every day [9]. Performing this recommended level of physical activity by WHO has been found to be challenging even for normal children. Hence, this recommended daily MVPA becomes much more difficult for C-DS as they are predisposed to obesity and physical inactivity [10]. As a result, initiatives need to be programmed in order to increase the quality and quantity of physical activity for C-DS. The WHO (2020) defines physical activity as action that uses energy generated by muscles. Physical activity has been found to be salubrious for cognitive, physical and psychological health [11]. However, most C-DS do not engage in the recommended MVPA, and therefore they are less likely to participate in more than one sport [12,13]. A previous report from KSA stated that C-DS display sedentary behaviors and participate in less physical activity than children without DS. Furthermore, in comparison to the control group, C-DS (aged 8–12) had a higher body mass index (BMI) and higher levels of physical inactivity [5]. It has been reported that interventions aimed at encouraging C-DS to exercise regularly and systematically in educational or extracurricular settings should use engaging, motivating activities to increase the likelihood that the target population will comply with the program. It is in this context that games and the many techniques they employ become crucial pedagogical tools for enhancing DS patients’ capacity for physical activity and movement. Moreover, a distinct but complementary perspective on motor games has been presented. The goal of this approach is to use motor games as a tool to prescribe exercise by quantifying and structuring them in various ways, such as a training program, with the hope of achieving specific goals connected to bioenergetic and metabolic demands and how these are related to health in terms of sport and motor features. Thus, this becomes not only a pedagogical and didactic resource but also a strategy for the adherence of individuals with DS and their families to physical activity [14].

The use of technology among children is increasing from day to day. A few studies have started to investigate the effect of technology use on children from various aspects [15,16]. In a research study [17], typically developing young children who were exposed to more than one hour of television per day were suspected of having delays in cognitive, language and motor development. Zamani (2009), previously reported that children’s attraction to electronics causes many problems related to mental, physical, and social development for C-DS, such as obesity, social isolation, the stimulation of anger and violence, epilepsy due to games and other physical and mental damage [18].

The increasing prevalence of technology use among children has led many researchers to study the effects of technology use on children. However, there are few and limited studies on the correlation between electronic technology use and the level of physical activity among children. In line with the literature survey carried out, it can be hypothesized that increased usage of digital-technology-based devices acts as a barrier to the recommended physical activity for the C-DS. Therefore, the purpose of this study was to investigate the correlation between the use of technology and the level of physical activity among C-DS.

## 2. Methodology

### 2.1. Design

A cross-sectional study was designed to collect the comprehensive data and semi-structured questionnaires were sent to the targeted sample using the internet. All the prerequisites of ethical considerations were consummated before carrying out the study. Post-approval from the Institutional Review Board (IRB Log number: 17-0198) of Princess Nourah bint Abdulrahman University, the survey was sent to three DS centers (Riyadh Specialized Rehabilitation Center, Down Syndrome Charitable Association (DSCA) Center and Efadh DS center) to be emailed to the mothers of C-DS in their list and to be posted on their social media.

### 2.2. Participants

A sample size of 90 mothers of C-DS (with children between 6–12 years of age, with mild to moderate intellectual disability [19]) were included in the study from three different centers across Riyadh, KSA; mothers of C-DS with severe to profound intellectual disability were excluded. There were only three DS disability centers in Riyadh at the time of recruiting participants for the presented study. The reason for limiting our sample size to 90 was based on the availability of these mothers to respond to the questionnaire and their consent to share their personal data.

### 2.3. Data Collection

After consenting to their participation in the study, all the mothers were requested to respond to all three sections of the semi-structured questionnaire, which included demographic variables, electronics usage behavior and different types of physical-activity-related questions in the Arabic language. The survey was completed by the mothers of C-DS, and it took approximately 8–10 min to complete. Incomplete questionnaires and those that were submitted twice were discarded. The data collection and analyses were conducted simultaneously and the questions that were semi-structured or open-ended were transcribed verbatim and analyzed by using the following steps of thematic analysis. In the first step, all the questionnaires that received a full response were reviewed by the authors and checked in order to ensure their accuracy. In the next step, to maintain the anonymity and confidentiality of the data, codes were given to each included participant. In the third step, initial coding was performed based on the responses of the mothers.

### 2.4. Measures Included in the Questionnaires

#### 2.4.1. Demographic Variables

Sociodemographic data were gathered using a questionnaire developed by the researchers. There were three sections; the first section contained information about the mother’s age, degree of education and marital status. The second section included information about the child’s age, gender, intellectual disability level and any medical conditions. The third section contained data on the total number of children in the household and the family’s monthly income.

#### 2.4.2. Children’s Physical Activity Questionnaire (CPAQ)

The CPAQ is a parent-reported questionnaire that assesses the type, frequency and duration of physical activity and sedentary behaviors over the preceding seven days, encompassing school and leisure time. The response options were indicated by whether the respondent engaged in a particular activity (“yes” or “no”) and by the amount of time spent on the activity during the week (minutes/week). The CPAQ also included information on the mode of transportation utilized to carry the child to and from day care. Along with sports, the length of leisure activities (such as walking or skating) was recorded. The questionnaire collected data on total non-school sedentary time, which covered activities, such as art and craft, homework, listening to music, traveling by car/bus, reading, watching television/videos and using the computer. The total screen time comprised all time spent outside of school hours on computers/internet/devices, watching television/videos and playing computer games. The CPAQ has a validity coefficient of 0.42 (*p* = 0.04) and a reliability coefficient of 0.39 (*p* < 0.05) for assessing moderate and vigorous physical activity via the questionnaire [20,21].

#### 2.4.3. Research Questionnaire to Evaluate the Impact of Technology Usage on C-DS

This is a parent-reported questionnaire that Dr. Jacqui Taylor of Bournemouth University’s Psychology Research Centre (School of DEC), Talbot Campus developed to investigate the impact of technology on children. It includes 24 questions about the child’s background, technology use, after-school activities, sleep patterns, behavior and emotions. This questionnaire is suitable to be used with children aged between four and twelve [22,23].

### 2.5. Translation of the Questionnaires

The CPAQ and Research Questionnaire on the Impact of Technology usage on Children was translated into Arabic and tested before being used, as advised by Bradley (1994) [24]. The surveys were initially translated from English to Arabic by a team of three different health professionals competent in both languages. Another expert blinded to the original English version of questionnaires reverse-translated them into English. In both cases, the backward-translated English versions of the surveys were evaluated by a professional simultaneous Arabic-English interpreter.

### 2.6. Pilot Study

In ten participants, the first-translated questionnaires were tested. The pilot test used self-administered questions. Later, the researcher interviewed participants in person to see if any questions were puzzling or difficult to answer. The two first translators amended the first-translated questionnaires to generate the final Arabic version based on comments and suggested adjustments from the pilot study. Five additional volunteers were used to re-pilot test the final Arabic version to confirm that the instructions, questions, and response options were understandable.

### 2.7. Data Analysis

The identity of the participants was encrypted to maintain confidentiality before running the statistics. IBM SPSS Statistics for Windows, Version 23.0, was used to analyze the data (IBM Corp., Armonk, NY, USA). Frequencies and percentages were used to represent general characteristics. Descriptive statistics (mean and standard deviations) were also presented. The presented sample was found to be slightly skewed and kurtotic for both male (skewness = −0.252 ± 0.472; kurtosis = −0.968 ± 0.918) and female (skewness = 0.194 ± 0.464; kurtosis = −1.321 ± 0.902), however, it did not differ significantly from normality. Hence, it was assumed that the presented sample was approximately normally distributed (Shapiro–Wilk test *p* value for males = 0.337 and for females = 0.096) in terms of skewness and kurtosis. The Pearson chi-square test was employed to examine the correlation between technology usage and physical activity level in C-DS. Spearman’s test was used to find out the association between technology usage (without watching television) and physical activity. A probability of *p* < 0.05 was considered to be statistically significant.

## 3. Results

### 3.1. Description about Mothers of C-DS

A total of 76 responses were received for the survey; 49 responses fit the inclusion criteria and 27 were excluded due to incomplete data or the fact that the child had a severe intellectual disability. As presented in Table 1, most of the responses were from mothers over the age of 45 (*N* = 22), which represented 44.9% of the total sample. Moreover, mothers aged 40–45 represented 28.6% (*N* = 14), followed by mothers aged 35–40 (*N* = 8) 16.3%, aged 30–35 (*N* = 4) 8.2% and mothers aged 25–30 (*N* = 1) 2%. Most of the mothers were married (*N* = 45), however, there were also separated (*N* = 2), widowed (*N* = 1) and divorced mothers as well (*N* = 1). Women with bachelor’s degrees (*N* = 26) accounted for 53.1% of mothers, while women without a high school diploma (*N* = 13) accounted for 26.5%, those with a postgraduate degree accounted for 12.2% (*N* = 6), and those without a high school diploma (*N* = 3) accounted for 6.1 %. Mothers with three children (*N* = 11) had the greatest percentage (22.4%), followed by six children (*N* = 10) (20.4%) and four children (*N* = 8) (16.3%). Families with five children (*N* = 7) comprised 14.3% of all families. Families with seven children (*N* = 6) comprised 12.2% of all families, 6.1 % of families had one child (*N* = 3), 4.1% had nine children (*N* = 2), and 2% had two or eight children (*N* = 1) (Table 1).

A family’s monthly income was between SAR 5000–10,000 for 38% of the participants, followed by families who earned more than SAR 20,000 (*N* = 12) with 24.5%, while an equal number of families had a monthly income ranging from SAR 10,000–15,000 (*N* = 6), SAR 15,000–20,000 (*N* = 6) and less than SAR 5000 (*N* = 6) (Table 1).

### 3.2. Description of C-DS

The participating C-DS were almost equal from both the genders, with 51% from mothers with female C-DS (*N* = 25), and 49% from mothers with a male child with DS (*N* = 24). The study targeted children between the ages of 6–12 years; the mean age was 8.55 and the S.D. was 2.14. The majority of the children (79.6%) were found to possess an IQ level ≥50 (*N* = 39) and the remaining 20.4% possessed an IQ level between 35–49 (*N* = 10) (Table 2).

### 3.3. Descriptive Information on the Child’s Technology Acquisition

The study found that 67.3% of the children possessed personal electronic devices (*N* = 33), whereas the remaining children used the devices of their family members (*N* = 16). The mean age when the children owned their first device was (6.09 ± 2.57). The maximum number of devices a child owned was three, the minimum was one. However, 93.9% of the children held only one device (Table 3).

### 3.4. Relationship between Technology Use and the Characteristics of the Mother

The Pearson chi-square test, which was used to examine the association between technology usage and maternal features, revealed that there was no significant correlation between technology use and maternal traits (age, marital status, family monthly income and the number of children in the family). Only between the mother’s education degree and the child holding a gadget was a significant link revealed (*χ^2^* = 0.862, DF = 4, *p* = 0.05). This was a positive association, indicating that the more educated the mother was, the more likely it was that the child would acquire a device.

### 3.5. Technology-Usage Pattern by C-DS

The hours spent on technology were determined using the Research Questionnaire on the Impact of Technology on Children. It categorizes the hours a youngster spends watching television each week into three categories: 1–10 h, 10–19 h or ≥20 h. The number of hours per week that a child spends playing on a portable game console, such as an iPad, iPod, PSP or Nintendo DS, as well as his or her use of social networking sites, was classified into four categories: Never <1–5 h, 6–10 h or ≥10 h.

The majority of the C-DS (81.6%; *N* = 40) watched TV for 1–10 h per week, while 18.4% (*N* = 9) watched for 10–20 h per week. In terms of time spent on a portable console, 46.9% (*N* = 23) spent 1–5 h per week, 18.4% (*N* = 9) spent 5–10 h per week, 14.3% (*N* = 7) spent more than 10 h per week and only 20.4% (*N* = 10) of C-DS did not play on a portable console. Additionally, Table 4 indicates that 87.8% (*N* = 43) of C-DS did not use social networking sites, whereas 4% (*N* = 4) did. 8.2% utilized social networking sites between 1 and 5 h per week, and youngsters who used social networking sites between 5 and 10 h per week (*N* = 1) or more than 10 h per week (*N* = 1) each account for 2%.

### 3.6. Evaluation of Physical Activity

A CPAQ tool was used to analyze the physical activities among the C-DS. A three-level classification of Metabolic Equivalent of Task (MET) was used; Light intensity activities with MET less than 3 h/week, Moderate intensity activities with MET from 3 to 6 h/week and Vigorous intensity activities with MET more than 6 h/week. The MET values of activities were acquired from the revised version of the Compendium of Physical Activities. Hence, every child’s physical activity in hours/week unit was estimated and converted to the matching MET value [25]. The maximum number of hours in every category was derived as the level of child’s physical activity.

Most physical activities undertaken by C-DS were classified as low level (MET ≤ 3) 21.25 ± 13.11 h per week, followed by moderate level (MET 3–6) 12.15 ± 8.56 h per week. The least-often performed activity was vigorous physical activity (MET > 6) 2.08 ± 3.24 h per week (Figure 1).

### 3.7. Relationship between Physical Activity and Gender of the Child with DS

The male C-DS (2.35 ± 3.96 h/week) were more likely to practice high level physical activities than female children (1.66 ± 2.29 h/week). On the other hand, female children practiced moderate- (13.41 ± 8.42 h/week) and low-level activities (22.56 ± 12.72 h/week) more than male children (Figure 2). Nevertheless, no significant correlation between gender of the child and the level of physical activity was observed (*χ^2^* = 0.001, DF = 1, *p* = 0.97).

### 3.8. Relationship between Physical Activity and Technology Use

Children who spent ≤5 h using technology (2.08 ± 3.24 h/week) performed vigorous activities, whereas children who spent ≥6 h using technology did not perform any vigorous activities. Similarly, children who were engaged for 5 h or less using technology (21.25 ± 13.11 h/week) performed low levels of physical activities more than children who spent 6 h or more using technology (13.85 ± 11.52 h/week). Contrastingly, children who spent 5 h or less using technology (12.15 ± 8.56 h/week) performed moderate level of activities less than the children who spent ≥6 h using technology (16.75 ± 1.76 h/week) (Figure 3).

The Pearson chi-square test was employed to assess the correlation between technology usage and physical activity. The outcomes reveal that there was no significant correlation between technology use (TV, portable game consoles and social networking sites) and the different levels of physical activity.

Moreover, Spearman’s test was exercised to examine the association between technology usage (without watching television) and physical activity and found no significant correlation between technology use (portable games consoles and social networking sites) and levels of high physical activity (*p* = 0.803), moderate physical activity (*p* = 0.142), or low physical activity (*p* = 0.432). Although the association was not statistically significant, a negative correlation between technology use and high-level activities (r = −0.037) was noticed. Contrastingly, a positive association between technology usage and moderate-level (r = 0.213) and low-level (r = 0.115) activities was observed.

## 4. Discussion

The emergence of digital technology has affected the children in both positive and negative ways. In one aspect, the children gain knowledge and advance in the use of digital technology at a very tender age, whereas on the other side, it affects their physical activity negatively. The situation is worsened when the child is physically disabled due to disorders like cerebral palsy and DS. The effect of technology on different levels of physical activity in young children has been studied previously in the typically developed child [26], autism disorder [15] and developmental delay disorders [17], however, no study has been reported in the DS population of Saudi Arabia. The current study’s major objective was to investigate the relationship between technology usage and levels of physical activity in C-DS in Riyadh region of Saudi Arabia. The study enrolled 49 mothers of C-DS who were between the ages of 6 and 12 years and had mild to moderate intellectual disability.

Mothers over the age of 45 responded at a higher rate than other age groups. According to some reports, older mothers (≥45 years) are more likely to have a child with DS than younger mothers [27]. Although there is lack of knowledge regarding the incidence of DS according to the mother’s age in Saudi Arabia, the outcomes of this study may provide us with indirect evidence about DS occurrence, nonetheless, additional research is necessary to validate this. This study found that mothers of female C-DS responded more positively. To our knowledge, no study has been reported so far in Saudi Arabia to determine the incidence of DS by gender, but another study indicated that DS is more prevalent in female than male children [28,29,30]. Additionally, the findings indicated that the majority of DS children began using electronic devices at the age of four, and the majority of them already owned a device. Numerous research has been conducted to determine the association between device ownership and screen time usage among youngsters [31,32,33]. These research conclusions corroborate our findings, which indicate that youngsters who own their gadgets are more likely to spend time on their electronic gadgets. This finding asserts that device ownership facilitates access to entertainment with different games, podcasts, programs, and similar other applications [31]. Despite the short sample size, the presented study found a positive association between the mother’s education level and the child holding a gadget. The better-educated the mother of a C-DS was, the more likely it was that the youngster would use technology. In contrast to these findings, Carson et al. (2012) stated a negative correlation between use of technology among normally developing children and their parents’ educational level [22,34].

This study found that majority of C-DS (81.6%) spent between 1–10 h/week watching TV. Previously it has been reported that normally developing children who watch TV excessively are more likely to develop health and developmental problems, such as obesity, delayed development of cognition and language [26,35,36], as well as motor developmental skills [17]. Such developmental complications may be exacerbated in children with neurological disabilities. The American Academy of Pediatrics recommends that children ≥2 years should not watch TV for more than two hours per day [37,38,39]; hence, C-DS should spend less time watching TV. In this study, 46.9% C-DS spent between 1–5 h/week using portable electronic gadgets. Additionally, the results show that 87.8% of all the C-DS in this study were not using social media, which could be due to the cognitive requirements for such task. Many studies indicate that C-DS have varying degrees of cognitive damage, with most of them classed as mild (IQ: 50–55–69) or moderate (30–35 to 50–55), while a minority (10%) are classified as severe (IQ: 20–25 to 30–35) [40,41]. A prior study of adolescents with attention deficit disorder (ADD) found that those with ADD watched more television and used social media less than those with other disabilities, such as intellectual disabilities, speech or language impairments and learning disabilities [42]. Similarly, it was found that children with ADD spend more time on television and video games than their typically developing siblings and less time using interactive video games or social media [43]. It is assumed that television entertainment and electronic virtual games are appealing to individuals who have difficulty, due to the nature of the challenges faced by them, in engaging in activities requiring cognitive and socializing abilities [42].

There was no significant correlation between technology usage and physical activity levels in our study. In addition, most of children’s physical activities were classified as low-level physical activities. In addition, there was no significant association between the child’s gender and his or her level of physical activity. The outcomes are coherent with the former reports that incorporated objective and subjective measures [44,45]. Although numerous studies [1,6,46] have revealed limitations in physical activity in C-DS, the limitations could be ascribed to a variety of different reasons. For instance, limitations may be related to co-occurring medical issues, as 50% of C-DS have congenital heart disease and 10–30% have atlanto-axial/atlanto-occipital instability [47,48]. Additionally, they have a poor cardiovascular fitness, limited muscle strength and a predisposition to obesity/overweight. The reason for the high rate of overweight/obesity in C-DS is multifaceted, since it may be a result of physiological, sociological, environmental, or psychological factors [49]. Physical activity restrictions, particularly in Saudi Arabia, could be attributed to the country’s environment. Saudi Arabia’s climate is defined by northward-moving winds that generate sand and dust storms. Numerous research has found that C-DS are more prone to infections of the ear, sinuses, and respiratory tract, including the throat, as well as consequences from these illnesses [50]. Additionally, few reports have linked health problems in general and respiratory problems to climate change, particularly for those who face everyday challenges with medications and mobility [51]. Another limiting factor that could explain the low level of physical activity in C-DS and their increased reliance on technology use is the scarcity of physical activity facilities designed for children with special needs [52].

## 5. Conclusions

The outcomes of this study reveal that C-DS began using electronic devices at an early age, and most of them had already owned one by the age of four. In contrast to previous reports, this study found a positive association between the mother’s education level and the child holding a gadget, which asserted that the higher the mother’s educational qualification, the more likely the C-DS would use electronic devices. Moreover, this study found that there was no significant relationship between technology use and the level of physical activity in C-DS. While that the manifestation of DS restricts participation of C-DS in physical activities, they should still be encouraged to engage in different types of physical activities that normal children are involved in. As a result, the recommended level of physical activity for disabled children is the same as that of those without disabilities. The physiological and psychological advantages of physical activity are proportional to the amount of activity undertaken and hence it will prove to be beneficial for the C-DS.

## 6. Limitations

The initial barrier in this study was a time constraint, as this project had to be completed in a limited time. Due to the time constraint, the sample was limited to the DS cases available in Riyadh, Saudi Arabia. Therefore, the findings of this study cannot be extrapolated to other circumstances or regions of the country. In addition, mothers who had no access to computers or the internet may have been excluded from the online questionnaire. Due to the high demand for childcare, the questionnaire’s length may have deterred mothers of C-DS from completing it.

## 7. Recommendations

Given these perceived barriers, family consultation programs should be established and adequately executed. Personalized physical activity programs for young adults with disabilities have been demonstrated to improve short- and long-term physical activity levels. By informing families about the importance of physical activity for their children’s health, healthcare professionals can help to reduce impediments to physical activities. Educational programs promoting the value of physical activity for C-DS should also be promoted. Environmental adjustments should also be sought to accommodate C-DS. Therefore, the Ministry of Education, Ministry of Social Affairs, Ministry of Health, and private investors should form well-organized partnerships to promote physical exercise among C-DS. Furthermore, for diverse age groups and diseases, the Ministry of Health should encourage health professionals to research over-reliance on technology.

## 8. Prospects of This Study

The results of our study highlight the need to raise awareness about the need for structured physical activity programs among C-DS, particularly through government and private sector-supported targeted programs. In regard to future research, it is proposed that this study be replicated in different locations of the KSA, so that findings can be compared to see whether barriers and facilitators vary by region. Last but not the least future research should also involve both parents in order to examine the differences in mothers’ and fathers’ perspectives. Finally, the involvement of both parents could also reveal whether physical activity educational programs should engage both parents or whether one parent is sufficient to care for C-DS.

## Figures and Tables

**Figure 1 ijerph-19-10958-f001:**
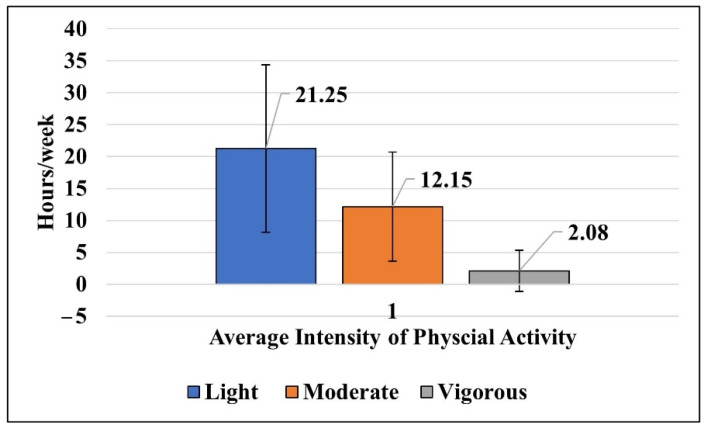
The means of light, moderate and vigorous activity hours for C-DS.

**Figure 2 ijerph-19-10958-f002:**
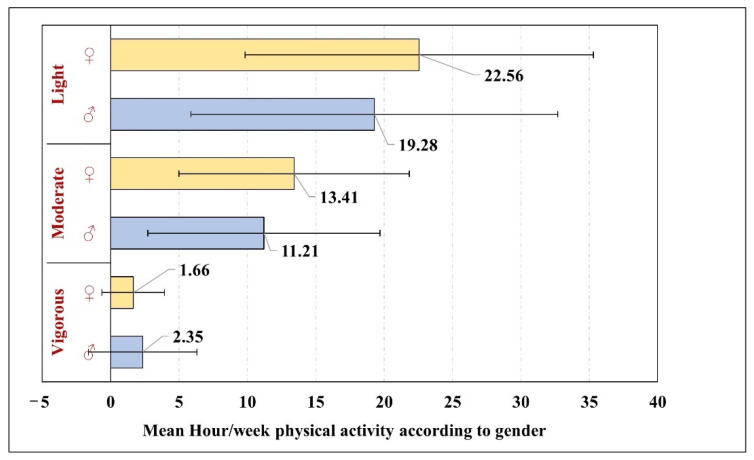
The means of light, moderate and vigorous activity hours for C-DS according to the child’s gender.

**Figure 3 ijerph-19-10958-f003:**
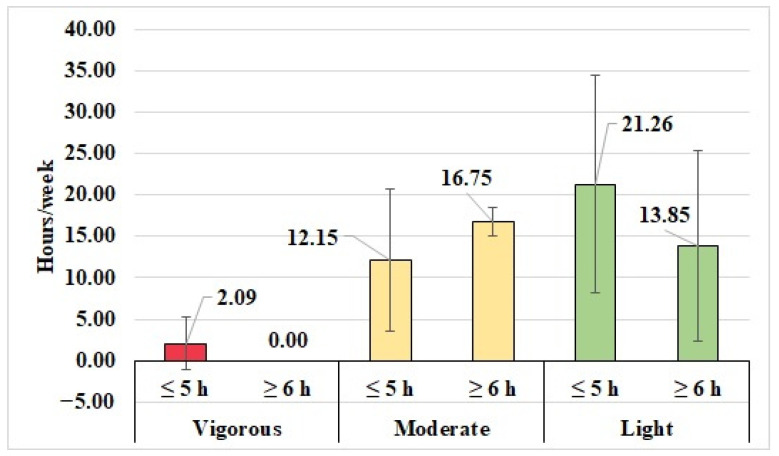
The means of light, moderate and vigorous activity hours for C-DS based on their technology use.

**Table 1 ijerph-19-10958-t001:** Descriptive information about mothers of C-DS.

Mother’s Age (Years)	Frequency	Percent (%)
25–30	1	2.0
31–35	4	8.2
36–40	8	16.3
41–45	14	28.6
above 45	22	44.9
Marital status	Frequency	Percent
Married	45	91.8
Widow	1	2.0
Divorce	1	2.0
Separated	2	4.1
Education level	Frequency	Percent
Not educated	1	2.0
Under high school	3	6.1
High school diploma	13	26.5
Bachelor’s degree	26	53.1
Postgraduate	6	12.2
Number of children in the family	Frequency	Percent
1	3	6.1
2	1	2.0
3	11	22.4
4	8	16.3
5	7	14.3
6	10	20.4
7	6	12.2
8	1	2.0
9	2	4.1
Family monthly income	Frequency	Percent
less than 5000	6	12.2
5000–10,000	19	38.8
10,000–15,000	6	12.2
15,000–20,000	6	12.2
more than 20,000	12	24.5

**Table 2 ijerph-19-10958-t002:** Descriptive information about C-DS.

Gender	Frequency	Percent
Male	24	49.0
Female	25	51.0
Age	Frequency	Percent
6	15	30.6
7	2	4.1
8	7	14.3
9	8	16.3
10	6	12.2
11	5	10.2
12	6	12.2
Mean Age= 8.55	Standard Deviation= 2.14	
IQ * Level	Frequency	Percent
IQ 50 or more	39	79.6
IQ between 49–35	10	20.4

* IQ: Intellectual Quotient.

**Table 3 ijerph-19-10958-t003:** Descriptive information on the child’s technology acquisition.

Does the Child Have an Electronic Device?	Frequency	Percent
Yes	33	67.3
No	16	32.7
Age of the child when he had his first device (years)	Frequency	Percent
2	2	6.3
3	1	3.1
4	9	28.1
5	4	12.5
6	3	9.4
7	2	6.3
8	5	15.6
9	2	6.3
10	3	9.4
12	1	3.1
Mean age for owning first device = 6.09 ± 2.57		
Number of devices	Frequency	Percent
1	31	93.9
3	2	6.1

**Table 4 ijerph-19-10958-t004:** Hours spent on Technology by C-DS.

Technology	Hours/Week	Frequency	Percent
TV	1–10	40	81.6
	10–20	9	18.4
Portable games console	0	10	20.4
	1–5	23	46.9
	5–10	9	18.4
	>10	7	14.3
Social networking sites	0	43	87.8
	1–5	4	8.2
	5–10	1	2.0
	>10	1	2.0

## Data Availability

The datasets used and/or analyzed during the current study are available from the corresponding authors on reasonable request.

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
