# Peer review of "Assessment of the Correlation between the Levels of Physical Activity and Technology Usage among Children with Down Syndrome in the Riyadh Region"

_ijerph, 2022, doi:10.3390/ijerph191710958_

Round 1

Reviewer 1 Report (Previous Reviewer 1)

The authors did a good job and corrected manuscript according to the comments. However, further revisions are required in the RESULTS section.

When the Pearson chi-square test was used for to examine the association, χ2 should be indicated in the Results section instead of t [now we see (t=0.862, DF=4, p=0.05); (t= 0.001, DF=1, p = 0.97)] because the Student's t-test is used to compare the means between two groups but not to examine the association.

 TO SUM UP I think the author(s) need to make the recommended corrections. 

Author Response

Honorable Reviewer,

The authors are grateful for your time and feedback. We think that all your comments were genuine and much needed. We appreciate your support and guidance in making the comments that helped us shape our manuscript in much better way.

In our capacity we have tried to reply to your comments satisfactorily. Please find the following content for the reply.

Response to Reviewer 1 Comments

Point 1: The authors did a good job and corrected manuscript according to the comments. However, further revisions are required in the RESULTS section.

Response 1: The authors would like to thank the reviewer for his valuable comments, the revisions were only possible because of his guidance.

Point 2: When the Pearson chi-square test was used for to examine the association, χ2 should be indicated in the Results section instead of t [now we see (t=0.862, DF=4, p=0.05); (t= 0.001, DF=1, p = 0.97)] because the Student's t-test is used to compare the means between two groups but not to examine the association.

Response 2: The authors apologize for the mistake and the correction has been made and highlighted in yellow.

Reviewer 2 Report (Previous Reviewer 2)

Good morning,
First of all, I would like to thank the editors of the journal for considering me as a reviewer of this paper. I would also like to thank the authors for the time they have taken to develop this interesting work. The following is a series of appreciations in order to be able to contribute some questions to improve the work if the authors consider adding considerations.

The summary contains the most important aspects of the work. In the introduction, it highlights the most important aspects that justify its study, although one of the pillars of the study is the use of technology. On the other hand, with regard to levels of physical activity, the justification is superficial, indicating the levels recommended by the WHO but not commenting on programs that could be beneficial for this type of population. We recommend reviewing the work of Farias-Valenzuela (https://scholar.google.cl/citations?user=l1OX6bQAAAAJ&hl=es).
On line 69 there is a citation error.

In the data collection, one of the paragraphs is shaded in red, so it comes with the change control included. The same occurs in the data analysis.
The instruments are well described and clearly identify the variables they measure.
Regarding the results, the first part corresponds to the characteristics of the sample. Therefore, it should appear in this section and not in the results. It is from point 3.3 onwards that it should appear in the results.
The exposition of the different results arranged in subsections is good and shows them clearly. However, the length of the results is too long. It will be shorter when you place the beginning in the corresponding section. Another option to consider is to provide only the significant and related results.
Regarding the discussion, it is recommended to highlight at the beginning the contribution of this work to science. Likewise, they should compare even more (because they already do so in a certain way) the results both with other studies with populations with SD and with studies that analyze the same variables in another type of population.
The conclusions of this work are clear, although they do not fully respond to the stated objectives and should be modified.

Author Response

Respected Reviewer,

The authors are grateful for your time and feedback. We think that all your comments were genuine and much needed. We appreciate your support and guidance in making the comments that helped us shape our manuscript in much better way.

In our capacity we have tried to reply to your comments satisfactorily. Please find the following content for the reply.

Point 1: In the introduction, it highlights the most important aspects that justify its study, although one of the pillars of the study is the use of technology. On the other hand, with regard to levels of physical activity, the justification is superficial, indicating the levels recommended by the WHO but not commenting on programs that could be beneficial for this type of population. We recommend reviewing the work of Farias-Valenzuela (https://scholar.google.cl/citations?user=l1OX6bQAAAAJ&hl=es). On line 69 there is a citation error.

Response 1: The authors thank the reviewer for his feedback, we identified and eliminated the deficiency. The authors are also grateful to the authors for providing the link of the article, it clarified the comment and gave us an insight of the gap. We have included the changes and rectified the citation error at line 69.

Point 2: In the data collection, one of the paragraphs is shaded in red, so it comes with the change control included. The same occurs in the data analysis. The instruments are well described and clearly identify the variables they measure.

Response: Authors thank the reviewer for his feedback and him noticing the error, the text color has been corrected to black.

Point 3: Regarding the results, the first part corresponds to the characteristics of the sample. Therefore, it should appear in this section and not in the results. It is from point 3.3 onwards that it should appear in the results.

Response: We would like to thank the reviewer for his valuable feedback. The authors would like to kindly inform the reviewer that since the study was aimed at finding “Relationship between and technology use and characteristics of the mother” the first part has been provided in the result section. The authors would like to seek more guidance from the respected reviewer, whether the reviewer assumes that the first part is unnecessary and should be removed or it should be placed in the methodology. In authors opinion, it is related to the study and is an illustration of the outcomes of our research, we put it in the result section.

Point 4: The exposition of the different results arranged in subsections is good and shows them clearly. However, the length of the results is too long. It will be shorter when you place the beginning in the corresponding section. Another option to consider is to provide only the significant and related results.

Response: The authors are grateful for the comment, the authors consider each section to be important as they are included in the interpretation of the outcomes. The authors are really puzzled on what section should be removed. Will the deletion of section 3.1 and 3.2 do the needful? Because we see the other sections in the results from 3.3 onwards to be directly related to the topic of discussion and objective of the study. Please we need further guidance on this issue.

Point 5: Regarding the discussion, it is recommended to highlight at the beginning the contribution of this work to science. Likewise, they should compare even more (because they already do so in a certain way) the results both with other studies with populations with SD and with studies that analyze the same variables in another type of population.

Response: The suggestion has been considered and possible changes have been made by authors. The authors agree to the reviewers’ point but still the authors think they have made sufficient comparison to make the objectives clear. Please suggest if you think it is necessary.

Point 6: The conclusions of this work are clear, although they do not fully respond to the stated objectives and should be modified.

Response: The conclusion has been revised and rewritten. Thank you so much for your valuable feedbacks and precious time.

The authors hope that the replies to the comments of our respected reviewer are satisfactory and fulfilling.

This manuscript is a resubmission of an earlier submission. The following is a list of the peer review reports and author responses from that submission.

Round 1

Reviewer 1 Report

 The topic of the manuscript is within the scope of the Journal and could be valuable to the scientific audience. The quality of the research design is acceptable.

 TITLE

The title of the article is accurate.

 ABSTRACT

Abstract reflects the work done and the conclusions drawn.

INTRODUCTION

Authors should formulate directional research hypothesis.

 METHOD

Some clarifications are however needed.

Please describe in details how purposive sampling technique was used for sample selection (Why three different centres across Riyadh? Why 90 mothers?).

Data analysis needs to be clarified. It is written that “The Pearson correlation coefficient was employed” but in the Results section we found that “The Pearson chi-square test was used to examine the association”.

The Spearman test should also be mentioned in the Data analysis section.

RESULTS

 The technique of data analyses seems appropriate but the results section is missing some important information. My suggestion is that all statistical notations should be written (not only p-values but also chi-square test values, degrees of freedom) because that is necessary for the proper interpretation of associations.

 DISCUSSION

 I suppose that not only limitations but also the future prospects (about further research) of the study must be defined and could be explained.

TO SUM UP I think the author(s) need to make the recommended corrections.

Author Response

Point 1: Authors should formulate directional research hypothesis.

Response 1: The directional research hypothesis was formulated and incorporated as recommended by the reviewer.

Point 2: Please describe in details how purposive sampling technique was used for sample selection (Why three different centres across Riyadh? Why 90 mothers?).

Response 2: The sought explanation has been included in the article under the section “Participants”.

Point 3: Data analysis needs to be clarified. It is written that “The Pearson correlation coefficient was employed” but in the Results section we found that “The Pearson chi-square test was used to examine the association”.

Response 3: The authors apologize for the confusion, the mistake has been rectified. Thank you for pointing out.

Point 4: The Spearman test should also be mentioned in the Data analysis section.

Response 4: The advise has been understood and implemented.

Point 5: The technique of data analyses seems appropriate but the results section is missing some important information. My suggestion is that all statistical notations should be written (not only p-values but also chi-square test values, degrees of freedom) because that is necessary for the proper interpretation of associations.

Response 5: We agree to the importance of the inclusion of data and hence the suggested statistical notations have been included by the authors in this article.

Point 6: I suppose that not only limitations but also the future prospects (about further research) of the study must be defined and could be explained.

Response 6: The authors have complied and added a section future prospects in the “Discussion” section.

Reviewer 2 Report

Thank you very much for considering me as a reviewer for this interesting paper. Thanks to the authors for their time. The following are a series of appreciations in order to provide ideas to improve the work.
The abstract is correct and the authors highlight the most important points.
The introduction begins in a general way to get to the most important variables and characteristics of this type of population. At the end of the introduction, the reader clearly knows which are the variables that are the object of this study and the objective is well defined on this basis.
Regarding the participants, it would be interesting to provide more data on the family. If there are siblings, the educational level of the mothers, the socioeconomic level of the families. These are considered necessary variables that can intervene in the results.
In the "data collection" section, they should indicate how the data were transcribed into the database.
It is very important that this type of research has the approval of an ethics committee and that it appears in the methodology section. If they have one, it should be included.
Regarding the statistical analysis, the authors should indicate if they have checked the normality of the sample, they do not indicate it.
Regarding the results, they are clear in their entirety, although it is considered that in order to publish in a journal of this level, more powerful statistical analyses should be performed to provide much more relevant results.
Regarding the discussion, the authors should not repeat the sample and age distribution, as they already do in the characteristics of the sample. The discussion is acceptable, although the authors repeat the data already presented in the results and should not proceed in this way.
As for the conclusions, it should be pointed out that this is not the place to refer to other works. The conclusions are drawn from the study in question and not from other studies. References to other works and the information they provide should be included in the introduction or discussion.

Thank you

Author Response

Response to Reviewer 2 Comments

Point 1: Regarding the participants, it would be interesting to provide more data on the family. If there are siblings, the educational level of the mothers, the socioeconomic level of the families. These are considered necessary variables that can intervene in the results.

Response 1: The authors agree to the points mentioned by the reviewer, the authors have provided the suggested data in table 1.

Point 2: In the "data collection" section, they should indicate how the data were transcribed into the database.

Response: The suggestion was considered by the authors and changes were made accordingly.

Point 3: It is very important that this type of research has the approval of an ethics committee and that it appears in the methodology section. If they have one, it should be included.

Response: The reviewer’s point has been noted and changes were made.

Point 4: Regarding the statistical analysis, the authors should indicate if they have checked the normality of the sample, they do not indicate it.

Response: The authors used SPSS latest version to analyze the data, and this software is automated to detect non-normal data.

Point 5: Regarding the results, they are clear in their entirety, although it is considered that in order to publish in a journal of this level, more powerful statistical analyses should be performed to provide much more relevant results.

Response: We have considered the recommendations.

Point 6: As for the conclusions, it should be pointed out that this is not the place to refer to other works. The conclusions are drawn from the study in question and not from other studies. References to other works and the information they provide should be included in the introduction or discussion.

Response: The suggestion was understood, and changes were made.

Round 2

Reviewer 2 Report

The authors have carried out some of the indications indicated, others have not been possible because there is an initial approach that marks the entire investigation and that cannot be modified. This means that the research is not of a sufficiently high level to be published in a journal of this category. Other suggestions could have been made but the proposed changes have not been observed with the depth that they should have been, for example, in the conclusions. It is recommended that if it is rejected, it should be published in other journals of a lower level.